# A database-based rather than a language model-based natural language processing method

## Abstract

Language models applied to NLP tasks take natural language as the direct modeling object. But we believe that natural language is essentially a way of encoding information, therefore, the object of study for natural language should be the information encoded in language, and the organizational and compositional structure of the information described in language. Based on this understanding, we propose a database-based natural language processing method that changes the modeling object from natural language to the information encoded in natural language. On this basis, the sentence generation task is transformed into read operations implemented on the database and some sentence encoding rules to be followed; The sentence understanding task is transformed into sentence decoding rules and a series of Boolean operations implemented on the database. Our method is closer to the information processing mechanism of the human brain and has excellent interpretability and scalability.

## 1 Introduction

Enabling machines to understand and use natural language as humans do is the ultimate goal of NLP. Many language models have been developed for related NLP tasks. For example: Word2Vec [2] and GloVe [3] models the correlations between words by constructing numerical representation of words (i.e., word vector) and expect to obtain a word-level understanding by computing the similarities between the word vectors. Seq2seq [6] and Transformer [7] are used for machine translation tasks, they model the mapping relations between words and the mapping relations between sentence structures in different languages. ELMo [4], GPT [5] and Bert [1] that pre-train language models on a large-scale corpus, are aimed at modeling the sequence features in corpus.

All these approaches of language models are modeling the surface features of language, while ignoring the fact that natural language is only a way of encoding information. We believe that the information described in natural language and the structural relations between these information do not change depending on the choice of different encoding methods. Therefore, we propose a database-based NLP method, which models the information represented by language and its organizational and compositional structure described in language, and provides methods for various NLP tasks, such as sentence generation and sentence understanding based on this model.

To summarize the contribution of this work:
• Our method changes the modeling object from language to the information represented by language, which makes the model we construct has excellent interpretability and scalability.
• We propose a brand new NLP approach that is different from rule-based and statistical model-based (i.e., language model) approaches, and it is more closer to the way the human brain processes information.

Submitted to 37th Conference on Neural Information Processing Systems (NeurIPS 2023). Do not distribute.

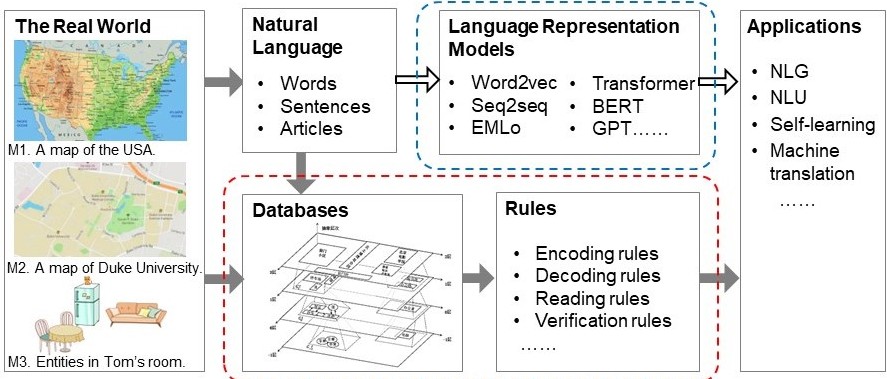

Figure 1: Differences in modeling objects. Language model-based methods take natural language as the object to construct the model. Database-based methods take both the information in the real world and the organizational and compositional structure of the information described in natural language as the objects to construct the model.

| | |
|---|---|
| The United States *is south of* Canada. | Duke University *is in* North Carolina. |
| The cat is *in* Tom's room. | The table *is in front of* the fridge. |
| The cat *is on top of* the fridge. | The sofa *is next to* the fridge. |

Table 1: Examples of sentences that describe the spatial position of target entities in the real world

- Our method directly confronts the challenge of "What is understanding ?" and "How to understand ?" and provides a convincing solution to the challenge.

## 2 Background

There are many kinds of information encoded in natural language, which need to be modeled and processed according to their different nature and characteristics. In this paper, we only take the spatial position information of entities as the object, and construct a model accordingly by learning how it is described and encoded in the language. People encode the spatial position information of entities in the real world into sentences, as shown in Table 1, and communicate them to each other.

Looking at the sentences in Table 1, we see that these sentences have the same structure: (Entity 1) + (...) + (Spatial relation) + (Entity 2), where "Entity 1" is the target entity whose spatial position we want to describe by the sentence, "Entity 2" is a helper entity that helps to locate the target entity, and "Spatial relation" describes the spatial relation between the target entity and the helper entity. As shown in Table 2 , there are three types of spatial relations commonly used in languages:1) spatial range relations, 2) spatial directional relations, and 3) spatial distance relations. the spatial directional relations can be further divided into 2.1) absolute directional relations and 2.2) relative directional relations according to the different reference systems.

The above findings in language reveal how people organize and store the spatial position information of entities in the real world. We can also see that people are used to using entities with a relatively stable spatial position as helper entities. We refer to entities with a relatively stable spatial position as immovable entities and entities whose spatial position is constantly changing as movable entities. The immovable entities and the spatial relations between them form a stable system that we will use to construct our model.

## 3 Model Architecture

We construct a tree-graph hybrid model to describe and store entities and the relative spatial relations between them. In a tree-graph hybrid model, the immovable entities are abstracted as square nodes and the movable entities are abstracted as round nodes. The spatial relations between the entities are abstracted as directed edges $E$. There are three steps to build our model:

| Spatial relations | | Lexical representations | Reference system |
|---|---|---|---|
| 1.Range relations | | **Inside**: in, at...
**Outside**: outside of... | (diagram: A, B, C) |
| 2. Directional relations | 2.1 Absolute directional relations | **East**: east of...
**West**: west of...
**North**: the north side of ...
**South**: the south side of... | (diagram: North/West/East/South, $E_a$) |
| | 2.2 Relative directional relations | **Top**: on, above, over, on top of...
**Bottom**: below, under, beneath...
**Left**: left of...
**Right**: the right side of...
**Front**: before, in front of...
**Back**: behind, back of... | (diagram: Top/Front/Left/Right/Back/Bottom, $E_r$) |
| 3.Distance relations | | by, near, next to, beside... | |

Table 2: Classification of the relative spatial relations between entities, and lexical representations of the relative spatial relations.

## 3.1 Tree Model

First, we use a tree model to describe the spatial range relations between entities. The spatial range relations $E_s$ is consist of two opposite directions, i.e., $E_s = \left\{ \overset{inside}{\longleftarrow}, \overset{outside}{\longrightarrow} \right\}$. For example, we use the tree in Figure 2 to describe the spatial range relations between entities "North Carolina", "Duke University", "Tom's room", "Classroom 15", "Table", "Sofa", "Fridge", "Cat", "Tom" and "Blackboard". The tree in Figure 2 can also be written in tabular form as shown in Table 3. In a tree, the child nodes with the same parent node should be spatially independent of each other, which means, there is no spatial range inclusion relation between them, if not, the child node must be moved up or down until all the child nodes are spatially independent of each other.

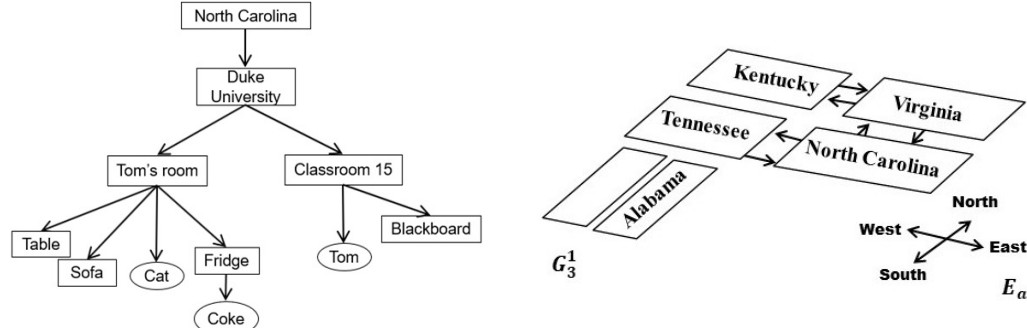

Figure 2: A tree that describe the spatial range relations between entities "North Carolina", "Duke University", "Tom's room", "Cat", etc.

Figure 3: A graph that describe the absolute spatial directional relations between some entities in M1 in Figure 1.

## 3.2 Graph Model

Then, We use graph models to describe the spatial directional relations between entities. The spatial directional relations can be future divided into 1) absolute directional relations $E_a$, which consists of four fixed directions, i.e., $E_a = \left\{ \overset{east}{\longrightarrow}, \overset{west}{\longrightarrow}, \overset{North}{\longrightarrow}, \overset{South}{\longrightarrow} \right\}$, and 2) relative directional relations $E_r$, which consists of six fixed directions, i.e., $E_r = \left\{ \overset{left}{\longrightarrow}, \overset{right}{\longrightarrow}, \overset{front}{\longrightarrow}, \overset{back}{\longrightarrow}, \overset{top}{\longrightarrow}, \overset{bottom}{\longrightarrow} \right\}$. Now, we can use the graph in Figure 3 to describe the absolute directional relations between some entities in M1 in Figure 1, and use the graph in Figure 4 to describe the relative directional relations between

| $E_s$ \ V | North Carolina | Duke University | Tom's room | Classroom 15 | Fridge |
|---|---|---|---|---|---|
| $\xrightarrow{inside}$ | Duke University | Tom's room, Classroom 15 | Table, Sofa, *Cat*, Fridge | Blackboard, *Tom* | *Coke* |
| $\xleftarrow{outside}$ | ∅ | North Carolina | Duke University | Duke University | Tom's room |

Table 3: The tabular form of the tree in Figure 2

| $E_a$ \ V | Kentucky | Virginia | Tennessee | North Carolina | Alabama |
|---|---|---|---|---|---|
| $\xrightarrow{east}$ | Virginia | ∅ | North Carolina | ∅ | ∅ |
| $\xrightarrow{west}$ | ∅ | Kentucky | ∅ | Tennessee | ∅ |
| $\xrightarrow{north}$ | ∅ | ∅ | Kentucky | Virginia | Tennessee |
| $\xrightarrow{south}$ | Tennessee | North Carolina | Alabama | ∅ | ∅ |

Table 5: The tabular form of the graph in Figure 3.

the entities in M3 in Figure 1. These two graphs can also be written in tabular forms as shown in Table 4 and Table 5.

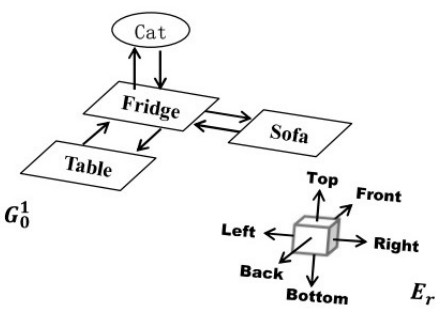

Figure 4: A graph to describe the relative spatial directional relations between entities in M3 in Figure 1

| $E_r$ \ V | Table | Fridge | Sofa |
|---|---|---|---|
| $\xrightarrow{left}$ | ∅ | ∅ | Fridge |
| $\xrightarrow{right}$ | ∅ | Sofa | ∅ |
| $\xrightarrow{front}$ | Fridge | ∅ | ∅ |
| $\xrightarrow{back}$ | ∅ | Table | ∅ |
| $\xrightarrow{top}$ | ∅ | *Cat* | |
| $\xrightarrow{bottom}$ | ∅ | ∅ | ∅ |

Table 4: The tabular form of the graph in Figure 4

### 3.3 Tree-graph Hybrid Model

At last, Take the nodes common to the tree and the graphs in Figures 2, 3 and 4 as connection points, then we can integrate the tree and graphs into a tree-graph hybrid model as shown in Figure 5. The tree-graph hybrid model describes spatial range relations between entities on the vertical structure (i.e., the inter-layer structure) and describes the spatial directional relations between entities on the horizontal structure (i.e., the intralayer structure). In a tree-graph hybrid model, the immovable entities and the stable spatial relations between them form a coordinate system, which can be used to locate the entities in the model (or database). A tree-graph hybrid model can be continuously extended upwards and downwards in the vertical structures to add new nodes, and continuously subdivided in the horizontal structure to add new nodes. Therefore, the tree-graph hybrid model could satisfy people's need to describe and store the spatial position of numerous entities in the real world. If the spatial position of one entity changes, just modified the related data accordingly in the model. In addition, we can also build datasets to store the spatial position information of movable

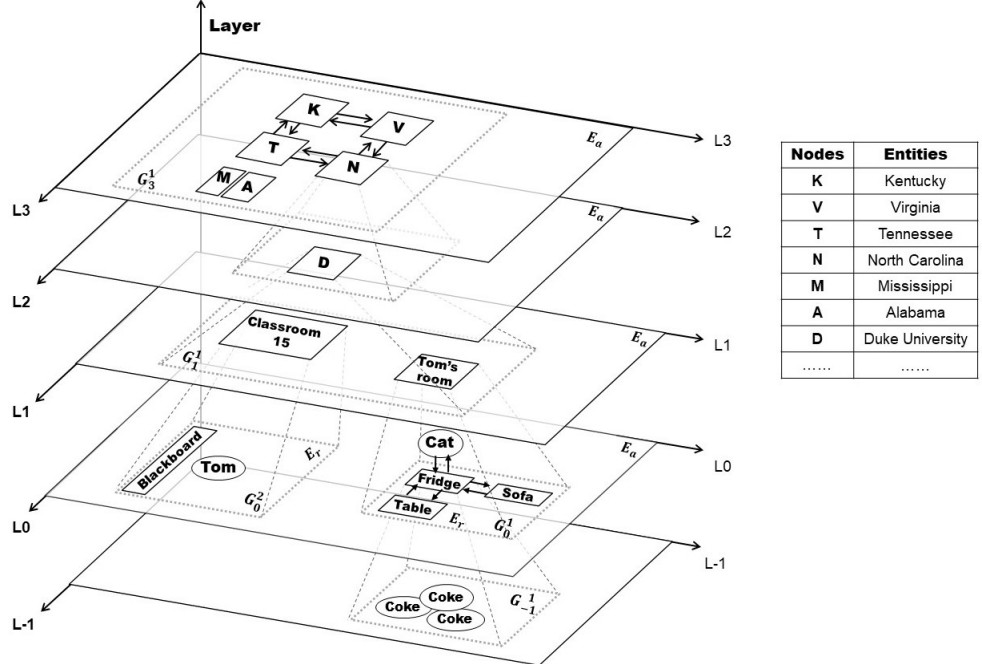

| Nodes | Entities |
|-------|----------|
| K | Kentucky |
| V | Virginia |
| T | Tennessee |
| N | North Carolina |
| M | Mississippi |
| A | Alabama |
| D | Duke University |
| ...... | ...... |

Figure 5: An example of a tree-graph hybrid model, which describes the spatial range relations between entities in the vertical structure (inter-layer structure), and describes the spatial directional relations between entities in the horizontal structure (intralayer structure)

entities to record their footprint. In deed, the tree-graph hybrid model build a information exchange bridge between language and the widely used numerical positioning systems, as shown in Figure 6.

Each layer of a tree-graph hybrid model can accommodate multiple subgraphs. Usually, the $E_a$ (absolute directional relations) is used as the reference system of the whole layer, and the $E_r$ (relative directional relations) is used as the reference system in each subgraph. As shown in Figure 5, the subgraph $G_0^2$ and $G_0^1$ take $E_r$ as their reference system, and the layer L0 is using $E_a$ as its reference system. In practice, when describing the spatial relation between two entities, lots of factors will affect our choice, such as the distance situation between the entities, the scale situation of these entities, etc. All of these factors can be summarized from practice and establish related rules, this part will be discussed in the following application section.

# 4  Application

Based on the tree-graph hybrid model, we can now generate a database to describe and store the spatial position of entities in the real world. This database can be used for many purposes. In this paper, we only present its use in NLG and NLU tasks.

## 4.1  Natural Language Generation

The purpose of the sentence generation task is to encode the information that needs to be conveyed into sentences. It consists of two subtasks: a) determining the information that needs to be conveyed and b) encoding that information into sentences.

### 4.1.1  Read Data From the Database

In this paper, the information to be conveyed is the spatial position of the target entity. Following the language expression, we will use a helper entity and the spatial relations between the helper and the target entity to describe the spatial position of the target entity. For example, if we want to describe the spatial position of the entity "Cat", we first need to find the corresponding node (target node) of the entity "Cat" in the database in Figure 5, then find the helper nodes that have a spatial relation with

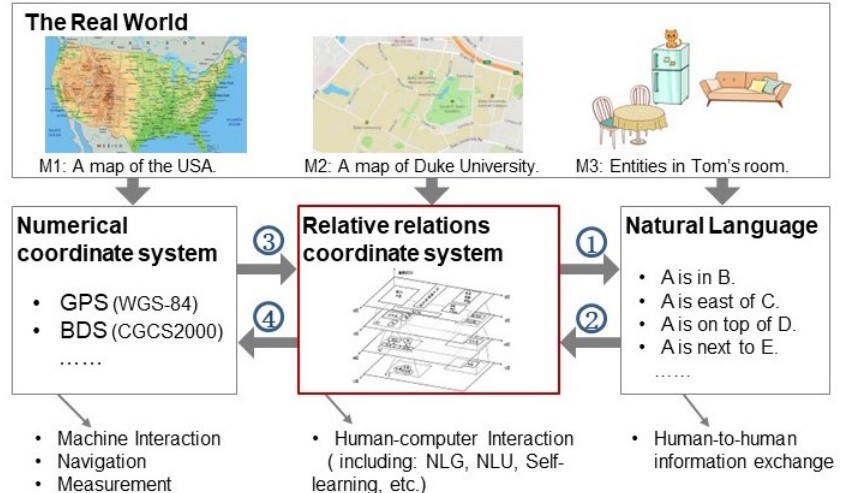

Figure 6: Three ways to describing (or encoding) the spatial position information of entities in the real world. And the routes of information exchange between different systems: ① sentences generation. ② sentences understanding.③ search for neighboring entities. ④ get the numerical position information of target entities.

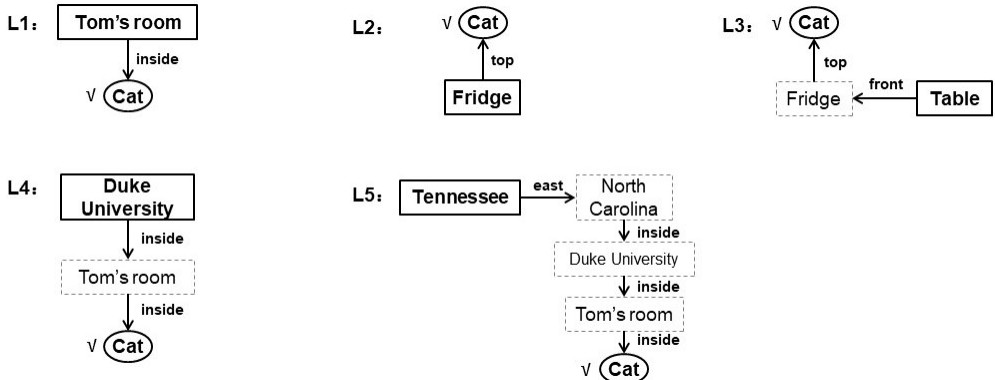

Figure 7: If we take the entity "Cat" as the target, then we can find the above 5 data chains in the database to help locate the entity "Cat". The nodes marked with "✓" are the target nodes.

the target node, such as the nodes "Tom's room", "Fridge", "Table", "Duke University", "Tennessee" and so on, then we can get 5 corresponding data chains as shown in Figure 7, which are composed of the target node, the helper node, and the spatial relations between them. Each of these 5 data chains can describe the spatial position of the entity "Cat", but their precision is different. If we sort these 5 data chains by precision, we can get the following result: $L2 > L3 > L1 > L4 > L5$. However, the precision is not the only goal we are pursuing. If we want to describe the spatial position of the entity "Cat" to a particular person, we also need to know how much this person knows about the spatial position of the 5 candidate helper nodes mentioned above, and what is the person's requirement for descriptive precision, so that we can filter out the appropriate one accordingly. Here, we will skip this part and go straight to the sentence encoding part.

### 4.1.2 Encoding Rules for Data Chain

Although the rules for encoding a data chain into a sentence vary slightly from language to language, but the following parts are mandatory: 1) the target and the helper nodes in a data chain, 2) the spatial relations between the target node and helper nodes in the data chain, and 3) the particular spatial correlation between the target node and helper nodes.

| Data Chain / Main parts | L1 | L1* | L2 | L2* |
|---|---|---|---|---|
| 1 Target node | The cat | The cat | The cat | The cat |
| 2 Helper node | Tom's room | Tom's room | the fridge | the fridge |
| 3 Spatial relations | in | *on top of* | on top of | *in* |
| 4 Particular spatial correlation | | | | |
| • True | is | | is | |
| • False | | *is not* | | *is not* |

Table 6: Examples of the mandatory parts for encoding a data chain.

| Data chain | Target node | Particular spatial correlation | Spatial relation | Helper node |
|---|---|---|---|---|
| L1 | The cat | is | in | Tom's room |
| L2 | The cat | is | on top of | the fridge |
| L3 | The cat | is | in front of ( next to) | the table |
| L4 | The cat | is | in | Duke University |
| L5 | The cat | is | on the ease side of | Tennessee |
| L1* | The cat | *is not* | *on top of* | Tom's room |
| L2* | The cat | *is not* | *in* | the fridge |

Table 7: Examples of sentence encoding for the data chains in Figure 7. All the above sentences are 100% correct, but some of them might be regarded as the right nonsense, and won't be adopted in practice due to their low precision in locating the target entity.

**Particular spatial correlation:** A data chain can describe both "true information" and "false information". Therefore, when encoding a data chain, speakers also need to give their opinion on whether the information described in the data chain is true or false. The speaker's opinion is described by a particular spatial correlation. For example, the particular spatial correlations that are listed in row 4 of Table 6 are the speaker's opinions on the information described in the data chains in Table 6.

**Operation rules for spatial relations:** If there is only one directed edge in a data chain, we can encode it directly, such as the data chains L1 and L2. If there is more than one directed edge in a data chain, e.g., the data chains L3, L4 and L5, we should first operate the directed edges in the data chain, then encode the result of the operation. Here, we summarize some operation rules as follows:

- Elimination operation: e.g., $\xrightarrow{inside} + \xleftarrow{outside} = \emptyset$, $\xrightarrow{left} + \xrightarrow{right} = \emptyset$, $\xrightarrow{north} + \xrightarrow{south} = \emptyset$...
- Union operation: e.g., $\xrightarrow{inside} + \xrightarrow{inside} = \xrightarrow{inside}$, $\xrightarrow{east} + \xrightarrow{east} + \xrightarrow{north} = \xrightarrow{northeast}$ ...
- Hybrid operation: when a data chain contains both spatial range relations and spatial directional relations, the relations in the upstream of the data chain is dominant, e.g., $\xrightarrow{inside} + \xrightarrow{top} = \xrightarrow{inside}$, $\xrightarrow{east} + \xrightarrow{inside} = \xrightarrow{east}$...

Applying the operation rules to the spatial relations on data chains L3, L4, and L5 yields the results below. Based on these operation results, we can encode the data chains L3, L4, and L5 into the sentences listed in Table 7.

- L3: $\xrightarrow{front} + \xrightarrow{top} = \xrightarrow{upfront}$;     L4: $\xrightarrow{inside} + \xrightarrow{inside} = \xrightarrow{inside}$;     L5: $\xrightarrow{east} + \xrightarrow{inside} *3 = \xrightarrow{east}$.

**Distance relations:** In some cases, e.g.: 1) the spatial distance between the target entity and the helper entity is very close, or 2) it is not necessary to provide the exact position of the target entity, then we can use the spatial distance relations as an alternative, just like the sentence L3 in Table 7

You may argue that the sentences we generated are too simple. However, at the initial stage of language appearance, it is just some simple words and short sentences. With the development of human beings, more and more information is encoded in language, then sophisticated words and long sentences emerged. Therefore, it is a good start to launch our research with some simple words and sentences.

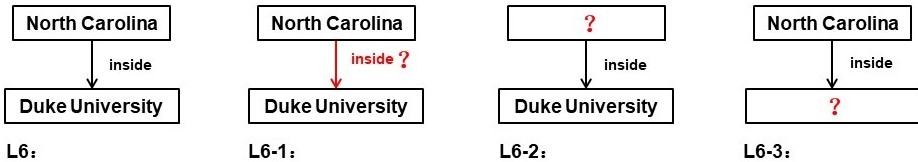

Figure 8: The data chain L6 and its three different cases.

| Data chain | Target node | | | | Helper node | |
|:---:|:---:|:---:|:---:|:---:|:---:|:---:|
| L6 | | Duke University | is | in | North Carolina | . |
| L6-1 | Is | Duke University | | in | North Carolina | ? |
| L6-2 | Which state is | Duke University | | in | | ? |
| L6-3 | | Which University | is | in | North Carolina | ? |

Table 8: Comparison of sentence structures that encode different information processing requests.

### 4.1.3 Encoding Rules for Processing Requests

Sentences encode not only the specific information to be conveyed, but also the processing requests for that information. According to the implicit processing requests in the sentences, we divided sentences into following three categories: 1) data description sentence (i.e., declarative sentence), 2) data verification sentence (i.e., the yes-no question sentence), 3) data searching sentence (i.e., WH-question sentence).

**Data description sentence:** The processing requirement implicit in a data description sentence is that listeners are expected to store the information in their databases. For example, teachers expect the students to remember what was taught in the class, and authors expect the readers to understand and remember the ideas shared in the book, and so on.

**Data verification sentence:** For a data verification sentence, listeners are expected to help verify whether the particular spatial correlations described in the sentence exist in their database, and then return the verification result as the response. For example, if speakers are not sure whether the spatial relation "inside" between the node (Duke University) and the node (North Carolina) exists, as shown in data chain L6-1 in Figure 8, they could express the processing request that ask listeners to help verity whether the "inside" edge exists by moving the word "Is" to the beginning of the sentence and adding a question mark at the end of the sentence, as shown in Table 8.

**Data searching sentence:** For a data searching sentence, listeners are expected to search for the missing information replaced by WH words in their databases and return the search result as the response. Take data chains L6-2 and L6-3 in Figure 8 as examples, speakers can use the word "which" to replace the missing parts and adjust the structure of the sentences, as shown in rows L6-2 and L6-3 in Table8, to express their expectation that the listener can help to search for the missing parts and return the search results.

## 4.2 Natural Language Understanding

In this paper, we only need to understand the spatial position information of the entities described in the sentences, the understanding of the other parts of the entities requires other databases, these databases will be published in other papers. The sentence understanding task consists of two parts: a) understanding of the processing requests of the specific information implicit in a sentence, and b) understanding of the specific information conveyed in the sentence.

### 4.2.1 Understanding of the Processing Requests

The specific processing requests are expressed by the specific sentence structures, specific feature words and specific punctuation. These can be used to classify the sentences and extract the processing requests accordingly.

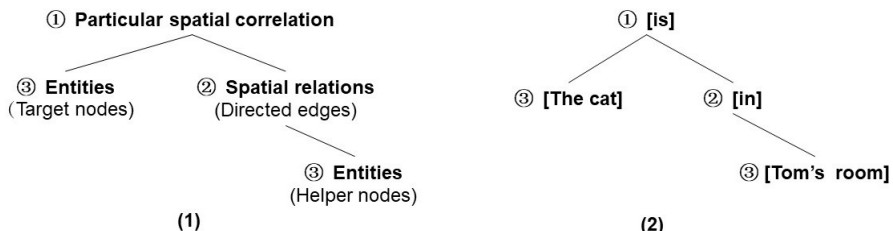

Figure 9: (1) General tree structure of sentences. (2) The sentence tree of sentence L1 in Table 7.

### 4.2.2 Understanding of the Specific Information

First, listeners need to chunk the sentence and extract components of the specific information. Considering the difference in the number of words and phrases used to represent each class of the components, the most efficient way is to chunk the sentences according to the order in Figure 9 (1). For example, listeners can chunk the sentence L1 in Table 7 into the sentence tree shown in Figure 9 (2). Then, listeners need to verify each parts of the sentence tree in their databases according to the flowchart shown in Figure 10. In the case of the sentence tree in Figure 9, listeners should first verity whether the helper entity "Tom's room" exists in their database, if the helper entity exists, go ahead; if not, it means that the listeners cannot get the position information of the target entity "The cat" through the helper entity "Tom's room", so the understanding mission fails. If the helper entity "Tom's room" exists, the listeners needs to verify whether the target entity "The cat" exists at the other end of the directed edge (i.e., "Inside" edge), if the target entity "The cat" exists, it means that the listeners understand the information conveyed in the sentence tree, although the information conveyed in the sentence tree is known to the listeners; If not, the listeners can create a target node at the other end of the "Inside" edge, to store the spatial position information of "The cat" in their database.

### 4.2.3 Responding to the Processing Requests

Strictly speaking, responding to the processing requests implicit in a sentence is not the sentence understanding task, but the sentence generation task. Here, we briefly introduce the responses to the different processing requests. **In a data description sentence**, the response is to store the specific information conveyed in the sentence, just as the step marked with a star in Figure 10. **For a data verification sentence**, the response is to return the verification results to the speakers. Take the sentence L6-1 in Table 8 as an example, in the listeners database, if the directed edge represented by the word "in" can be found between the node "Duke University" and node "North Carolina", the listener can reply "Yes, it is" as feedback to the speaker. If not, the listener can reply "No, it is not" as feedback to the speaker. **For a data searching sentence**, the response is to return the searching results to the speakers. Take the sentence L6-2 in Table 8 as an example, in the listener's database, if there is a node on the other end of the directed edge (which represented by the word "in"), the search mission succeeded, and the listener can give the lexical representation of that node to the speaker. If not, the listener can reply "I don't know" or "I don't have a clue" to the speaker, to let him or her know that the search mission failed.

## 5 Conclusion

We demonstrate the feasibility of the database-based method for NLG and NLU tasks, which takes information encoded in natural language as the object of study. So, what exactly are we study about natural language? As we have learned in neuroscience, humans receive information through neural pathways such as eyes, ears, mouth, nose, etc., and then send this received information to the brain for hierarchical processing and storage. Although we cannot explore how this information is processed and stored in human brains, but a small proportion of this information is encoded as natural language for external output. Thus, by studying natural language, we can investigate the mechanisms by which information is stored and processed in the human brain.

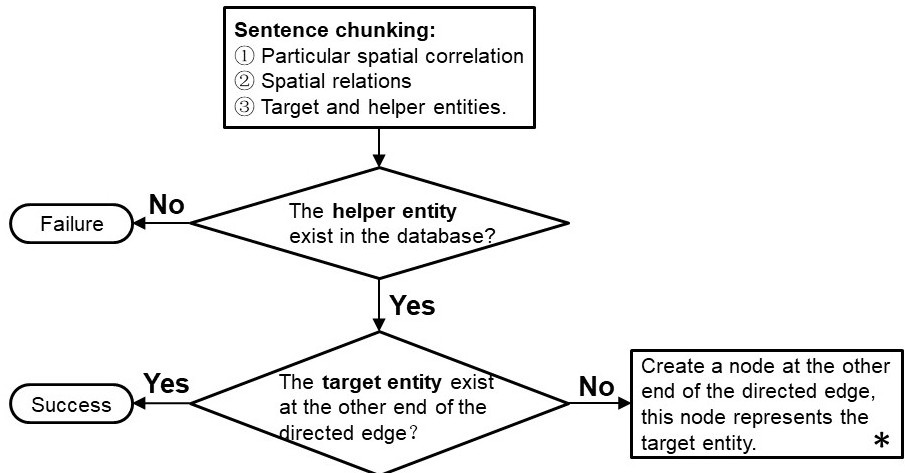

Figure 10: Understnading flowchart of data description sentences.

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
