# OpenReview forum: "A database-based rather than a language model-based natural language processing method"
_NeurIPS.cc/2023/Conference — Submitted to NeurIPS 2023_

### Official Review · Reviewer_s4Sj · 2023-06-29

**Soundness:** 2 fair
**Presentation:** 2 fair
**Contribution:** 1 poor
**Rating:** 1
**Confidence:** 5

**Summary:**

This paper proposes a new method for natural language processing. Instead of using language corpus, authors suggest to use database. Sentence generation is a linear schematization of a database-based representation. It is indeed an interesting idea.

**Strengths:**

Authors propose a brand new NLP approach that is more closer to the way the human brain processes information, and very likely on the way to a brand new neural process for NLP.

**Weaknesses:**

There is no experiment described in the paper. Authors shall finish experiments, then submit papers.

**Questions:**

Could we understand database as a base of data, which includes natural language descriptions?

**Limitations:**

Please read papers about mental spatial models, to see how simple spatial descriptions are generated from mental model.
For example, why most people say, San Diego is located further west of Reno? What could be the structure of the 'spatial database' in our neural mind?

---

> ### Author Rebuttal · Authors · 2023-08-10
>
> **Q11: Database (R4)**
>
> The “database(s)” is used to store information (knowledge). It is equals to the memory of the human being. The database factually describes the information (knowledge) in the real world and how they are stored and organized in the human brain. Please refer to the reply of Q2.
>
> **Q12: “Why most people say, San Diego is located further west of Reno? ” (R4)**
>
> The general explanation is that the memory in the human brain does not always reflect the real world 100%; it may contain some wrong information or confusion, such as, you may have misremembered the discussion deadline, but the memory can be corrected. Similarly, the model (or database) in the proposed method can also be modified.
>
> For the specific spatial relation model of "San Diego" and "Reno", sorry, I don’t know these two entities, the possible explanation is that:  1)“San Diego” or “Reno” has been listed in an improper layer (refer to the layer in Figure 5), this case and its solution have been discussed in lines 69\~72; 2) there is more than one directed edge between “San Diego” and “Reno” (refer to the L5 in Figure 7), the error occurs during the derivation, i.e., the derivation demonstrated in lines 139~ 151. No worries; everything can be rewritten.
>
> **Q13: “Read papers about the mental spatial models” (R4)**
>
> During my research, I did extrapolate the new paradigm to other related disciplines, such as classification problems, psychology, philosophy, etc. As far as the results are concerned, the new paradigm not only brings a completely new interpretation to the above disciplines or problems, but also provides some ingenious solutions.

---

### Official Review · Reviewer_zcb5 · 2023-07-01

**Soundness:** 1 poor
**Presentation:** 3 good
**Contribution:** 1 poor
**Rating:** 1
**Confidence:** 5

**Summary:**

This paper aims to take a novel standpoint with respect to all the neural network architectures working on natural language (NN-NL). According to the authors, these NN-NLs work on the surface of the language, disregarding that the language is encoding information. In their opinion, information should be represented as entities and relations as in databases. Consequently, the paper proposes NN architectures working on trees and working on different levels of information encoding (Fugure 5). Clearly, relations have properties such as transitivity that should be used during training and inference.


**Strengths:**


- The paper seems to propose a revolutionary way of thinking to neural networks


**Weaknesses:**

- The idea of information is a little weird. First, in this paper, the term information stands for structured information, and, apparently, the overall idea is to use structural information as the native form of encoding information as in databases. This is a little strange as there are not very large corpora of structured information. Indeed, having a large corpus is the key of success for these neural networks.

- The paper does not mention that the overall field of NLP was devoted to take Natural Language to a structured representation. From there, tasks were solved, eventually going back to natural language for specific tasks like dialog, question answering, and so on. This structured representation is at different levels: morphology, syntax, semantics, and, sometimes, pragmatics (e.g., in the form of speech acts). Only in the last decades NLP has pushed tasks from NL to NL (e.g., natural language inference, dialog) with architectures based on Machine Learning, which may be agnostic with respect to the structured representation of language utterances.

- Structured information mentioned in the paper may be correlated with the semantic representation of natural language. This is not even mentioned.

- There is a large body of studies on probing transformers to see if they replicate an NLP pipeline and to understand how they encode syntax, semantics and so on

- The basic idea of NN is to encode text, which may be retrieved and used by means of other similar text. This is encoding "information."

- The paper does not propose a dataset to work with.

- The paper has no experimental section






**Questions:**

See weaknesses

---

> ### Author Rebuttal · Authors · 2023-08-09
>
> **Q6: Large corpus and Neural network (NN) (R3)**
>
> Large corpus are the sample sets for training NN models; the NN models are a statistical inference model. See the reply in Q1; the statistical inference models are unsuitable for NLG and NLU problems. _The research object of my work is the information encoded in language, and the organizational and compositional structure of the information described in languages (lines 5\~7; lines 26~28),_ which differs significantly from the study object of the NN models-based methods. They are two different research methods with two other research objects. Therefore, corpus and NN are not mentioned in my work.
>
> **Q7: "Taking Natural Language to a structured representation." (R3)**
>
> I beg to differ, and my work is devoted to studying and revealing the structure of how information is organized and stored in the human brain. _Due to a small proportion of information in the human brain being encoded as natural languages for external output(lines 227~232),_ thus, we can take natural language as the $\underline{\text{medium of study}}$, but not the ultima $\underline{\text{research object}}$.
>
> **Q8: "The basic idea of Neural Network (NN) is to encode text." (R3)**
>
> I agree that the basic idea of NN is to encode text. _However, I believe that the ultimate goal of NLP is enabling machines to understand and use natural language as humans do (line 15)_. And I will propose a brand-new neural network in another paper or my book. _The new neural network_ is not based on the _chain rule_, and its working mechanism is more similar to the neurons in the human brain.
>
> **Q9: "The structured representation is at different levels: morphology, syntax, semantics, and, sometimes, pragmatics" (R3)**
>
> All the concepts or methodologies mentioned above are derived and developed to study NLP problems.  According to the first principles, after rediscovering and reconceptualizing natural language, _i.e., natural language is essentially a way of encoding information (lines 2\~3, line 24), and sentences encode not only the specific information to be conveyed, but also the processing requests for that information (lines 161\~165, lines 190\~192); The sentence understanding task consists of two parts: a) understanding of the processing requests of the specific information implicit in a sentence, and b)understanding of the specific information conveyed in the sentence (lines 186\~188)._ Naturally, the research methodologies will be adjusted and modified accordingly.
>
> **Q10: Lack of dataset to work with (R3)**
>
> See the reply in Q1.
>
> ------------------------------
>
> Special thanks to you for helping me review the previous work on NLP, so that I can expand my knowledge and think more deeply in the comparison process. As the saying goes: "If you don't pull out the light, you can't understand."

---

> > ### Author Response · Authors · 2023-08-18
> >
> > Hi, it was nice to discuss this with you. And I know it is hard to think in a completely new paradigm, and I have almost completed all the basic model parts, which gave me enough confidence in my work. Thus, whether the article is accepted or not, the work that follows will not be affected.
> >
> >  So is there anything you'd like to discuss that interests you?

---

### Official Review · Reviewer_fkGC · 2023-07-07

**Soundness:** 1 poor
**Presentation:** 2 fair
**Contribution:** 2 fair
**Rating:** 4
**Confidence:** 4

**Summary:**

This paper studies a database-based natural language processing method and proposes a tree-graph hybrid model based on three types of spatial relations. The model is further applied to both natural language generation and natural language understanding tasks.

**Strengths:**

The insight of borrowing human cognition to develop a natural language model processing method is worth exploring.

**Weaknesses:**

This paper lacks experimental observations and analyses to validate the effectiveness of the proposed method or support the conclusion.

**Questions:**

1. Capitalization typo in the title, line 9, and line 49.
2. Adequate literature review is needed.


**Limitations:**

Experimental results and analyses are needed to validate the effectiveness of the proposed method.

---

> ### Author Rebuttal · Authors · 2023-08-09
>
> **Q4: Typos (R2)**
>
> Thanks for your kind reminder; I will thoroughly check and fix them in the revised version.
>
> **Q5: Literature review (R2)**
>
> I tried to find some literature at the beginning of my work, but unfortunately, it was unavailable. After that, I throw myself into my work.

---

### Official Review · Reviewer_QFi3 · 2023-07-08

**Soundness:** 2 fair
**Presentation:** 3 good
**Contribution:** 1 poor
**Rating:** 3
**Confidence:** 4

**Summary:**

This paper advocates the separation of language and knowledge. It proposes a database-NLP method. The knowledge is contained in one or more databases whose internal structures can be a tree, a graph, or a hybrid. There are associated methods to query and retrieve from the database(s) the information. Finally, some "chain" based NLP methods connect the various tasks with the databases. In this paper, the database holds spatial relationship among various locations (e.g. Duck University is a location, Tennessee is a location, "the fridge" and "Tom's room" are locations). Overall, this reviewer feels that the approach proposed in this paper is very close to known art.

**Strengths:**

This approach has a very focused usage scenario. It can be used in a situation where "hallucination by modern language model" is not permitted.

**Weaknesses:**

Overall, this reviewer feels that the approach proposed in this paper is very close to known art.

**Questions:**

This reviewer does not have any questions to the authors.

**Limitations:**

The authors clearly presented the limitations.

---

### Author Rebuttal · Authors · 2023-08-07

Thank you for reading and commenting on my work. I look forward to more opportunities to communicate in the future.

**Q1: Lack of Experimental Section (All)**

I apologize for the operating error in the checklist on the initial submission. For the experiments option, I selected “yes”; actually, it should be a “n/a”. The experimental method is not a suitable option to verify the effectiveness of the proposed method for the following reasons.

1.	The proposed model is a data structure, not a statistical inference model. What describe in the proposed model are facts. Therefore, there is no need to verify its validity experimentally.
2.	Experimentation is not the only optional research method. Experimental methods are primarily used in the _natural sciences_, such as physics and biology. Logical testing methods are more popular in _formal sciences_ such as mathematics, logic, basic computing science, and other disciplines where artificial concepts are studied. The logical testing of the proposed method is listed in a separate item.
3.	In the areas of NLU and NLG, experimental results derived from _statistical inference models_ (e.g., language representation models) are not reliable; see proofs below:

> `Proof1:` The key assumption must be met to ensure language representation models work: $\underline{\text{understanding depends on context}}$. However, the assumption is untenable and easily disproved.\
&emsp; *Experiment:*  To understand the word "egg" without context.\
&emsp; *Process:*  We can understand the word "egg" by searching for the relative information (knowledge) of the entity "egg" in our memory (databases). For example: "color", "shape" and "taste" of eggs, "the texture of the eggshell", "eggs as food", "the relation between eggs and chickens", "some relevant scenes", etc. All the recalled information(knowledge) in our memory makes up our understanding of the word "egg". As shown in Figure A, if we know nothing about the entity" egg", the word "egg" is not understandable; by contrast, the drawing" egg" provides more clues to understanding.\
&emsp; *Result:*  It is easy to see that language understanding does not depend on context but relies on the relevant information (knowledge) in memory (database). Furthermore, the level of understanding of the word "egg" depends on the amount of relevant information (knowledge) in the memory (database). It won't exceed the scope of the memory (database). Note that only a fraction of the information (knowledge) involved in the understanding process (the data processing process) is encoded in language, forming the collectable samples.

> `Proof2:` Unbiased sample sets (corpus) are not available in the real world, which leads to the unreliability of statistical inference models trained on biased sample sets.\
&emsp; Language is a tool used to exchange information (knowledge) between people, who tend to encode and transmit information (knowledge) that the other parties do not know in language. Information (knowledge) shared by both parties is rarely encoded in language. Moreover, due to the limitation of language as an encoding tool, some of the information (knowledge) involved in the process of understanding cannot be encoded in language. Therefore, the above selective tendencies and the tool's limitation have resulted in an overall bias of language as samples (see Figure B).


**Q2: The reliability verification of the proposed method (All)**

Let's look at the following concepts from a different perspective. Then the logical relationships between them will be self-evident.

`Axiom：` The model proposed in Figure 5 is actually an axiom that factually describes the spatial relations between entities in the real world. An axiom does not need to be proved.

`Proposition:`  Sentences read from the model (or database) are the propositions.

The proposed NLG method is to derive propositions from the axiom. For example, all the sentences in Table 7 are propositions derived from the model in Figure 5.

The proposed NLU method is to verify propositions with the axiom. For example, all the sentences in Table 7 are verified as true. The sentence "The cat is in the fridge" is verified as false.

In the NLU process, if a given proposition is known to be true, the new information (knowledge) brought by the proposition can be written into the model (or database), further expanding the axiom. This is a self-learning process.



**Q3: Naming issue of the research object (R3)**

Thanks for raising the concern. Yes, it is crucial to clearly and accurately define and describe the research object in our work. In this paper, I use "information" to name the research object, which is only a compromised option. The other candidates are "knowledge" and "data" (see Figure C), but the things encoded or represented by the words "knowledge" and "data" cannot cover all that is involved in the NLG and NLU processes. After careful consideration, "information" is chosen. I know that "information" is a less clear-cut concept, and this issue may need to be discussed within the field. I will replace the "information" with "information (knowledge)" in the updated version temporarily.

---

> ### Author Response · Authors · 2023-08-18
>
> Knock, knock！ Did my previous reply cover all your concerns? Is there anything else we need to discuss? Any questions are fine.
>
>  I look forward to hearing from all of you as we still have some time left.
>
> Best regards!

---

> > ### Comment · Reviewer_s4Sj · 2023-08-18
> >
> > I like your intuition and nice ideas, and also agree with the limitations of language models. But, your current writing does not fit with NeurIPS (that is about neural computing). Would you argue for novel neural methods for determinate knowledge representation and reasoning?

---

> > > ### Author Response · Authors · 2023-08-18
> > >
> > > **Q14：Not fit with NeurIPS. (R4)**
> > >
> > > I’m not sure that I understood you correctly. Did you mean that the non-statistical inferential modeling class of methods does not fit NeurIPS? Since neural computing is an iterative algorithm-based statistical inferential class of method.
> > >
> > > My understanding is that the $\underline{\text{Neural Information Processing Systems (NeurIPS) is an international conference on machine learning and computational neuroscience}}$. Regarding the machine learning part, the proposed method has fully demonstrated its advantages in machine learning, i.e., we can give the sentences that describe the spatial information of entities in the real world to the model (Sentences of the same type as in Table 7). It can understand、learn and construct the database by itself—the detailed process is shown in Figure 10. For the computational neuroscience part, I recommend you read the book ”Neuroscience: Exploring the Brain” by Mark F. Bear; you might gain the same inspiration as I do. Humans receive information through neural pathways such as eyes, ears, mouth, nose, etc. All the received information is sent to neurons for computing, storage, and processing, and all the above activities of neurons make up the thinking activities of humans.
> > >
> > > Overall, I do not think my paper does not fit with NeurIPS.
> > >
> > > **Q15：” Would you argue for novel neural methods for determinate knowledge representation and reasoning?” (R4)**
> > >
> > > Inductive reasoning is one of the most critical methods of human thought. Human beings perceive both determinate and indeterminate knowledge from the environment. As for the determinate information (knowledge), people store and use it in their brains; and indeterminate knowledge is the object of scientific research.
> > >
> > > So, my answer is YES.
> > >
> > > Hope you have a nice day.

---

> > > > ### Comment · Reviewer_s4Sj · 2023-08-18
> > > >
> > > > Maybe the following paper can help you sharpen your methodology and tighten the connection to the neural community.
> > > >
> > > > Goyal, A.; and Bengio, Y. 2022. Inductive biases for deep learning of higher-level cognition. Proceedings of the Royal Society A: Mathematical, Physical and Engineering Sciences, 478

---

> > > > > ### Author Response · Authors · 2023-08-18
> > > > >
> > > > > **Q16:**  Thanks for your recommendation. However, I still believe that scientific research should not be limited to specific methods, and that solving problems is our ultimate goal. Thus, if the proposed method is indeed valid, its reliability is proven, and the results can be reproduced. Then, it is worth exploring.
> > > > >
> > > > > **Compared to selecting an appropriate method,  clearly recognizing the research object and the essence of the problem is more important.**

---

> > > > ### Author Response · Authors · 2023-08-18
> > > >
> > > > Follow up on Q14 and Q15
> > > >
> > > > **Q14:** After careful consideration again, I still believe that scientific research should not be limited to specific methods, and that solving problems is our ultimate goal. Thus, if the proposed method is indeed valid, its reliability is proven, and the results can be reproduced. Then, I think the proposed methods fit with NeurIPS.
> > > >
> > > > Not to mention, the fascinating chain of chemical reactions that a completely new way of thinking and solving a problem can bring about.
> > > >
> > > > **Q15:** Does the “novel neural method” refer to the new neural network mentioned in the reply of Q8 and its application on the proposed model (or database)? If so. Its working mechanism can be understood in such a way that the reasoning process is actually structurally represented in the proposed model (or database), and the reasoning process in the model (or database) is transformed into the activation process of the units in the database. Eventually, the reasoning result can be encoded in language for output. It can also be retained in the brain as a result of thinking. In short, encoding the result of reasoning into a language for output is not a mandatory option.
> > > >
> > > > Both the structures in the model (or database ) and in sentences are doubly encoded. The first layer structure in both model and sentences are the natural structure of the determinate information (knowledge). In sentence structures, the second layer is the processing request for specific information to be conveyed (lines161~165); and the structure of the model (or database) has encoded the reasoning path.
> > > >
> > > > Ok, the above part is kind of hard to understand. I am trying to explain that the novel neural is used to execute the reasoning in the proposed model (or database). I need to figure out how to demonstrate it in the future clearly.
> > > >
> > > > When you dig deeper, you'll see that NLU, NLG, and classification problems are all coding problems at heart.

---

> > > > > ### Author Response · Authors · 2023-08-20
> > > > >
> > > > > **Q17: Determinate knowledge**
> > > > >
> > > > > I would like to discuss this concept further. As we all know, there are _information gaps_ between people; for example, you (R4) know the spatial information of "San Diego" and "Reno," but I don't; that's an information gap. So for you, the spatial information is _determinate knowledge_, and for me, it is _indeterminate_. But you can encode this spatial information in language and pass it on to me, and I can learn this new information from you.
> > > > >
> > > > > From this point of view, people use natural language to reduce the information gaps between each other. Or we can say that language is a tool that allows us to copy other people's databases (memory) directly. Without such a tool, people have to learn from practice, a process that is extremely slow and inefficient. Do you know how exciting this is? It's like we can just copy the amount of money in someone's bank account without any actual labor. In this way, language is indeed the critical tool that has helped humanity to win overwhelmingly in the race for the survival of all living things on Earth.
> > > > >
> > > > > See, that's why I love discussions; it always inspires me to _rethink concepts I thought I knew_, and in turn, I'm able to gain a deeper understanding and thus get closer to the essence of things.
> > > > >
> > > > > Best wishes to all of you.

---

> > > > > > ### Comment · Reviewer_s4Sj · 2023-08-20
> > > > > >
> > > > > > Do not worry, most people do not have the information in mind about the relation between San Diego and Reno. What they know, I believe you too, are that San Diego is located in California and that Reno is in Nevada and California is located further west to Nevada. When they are asked the relation between San Diego and Reno, they search their mind, and have the above three pieces of related information, then they construct a 'mental spatial model', and inspect this model to conclude the relation between San Diego and Reno. (In this aspect, your databased perspective is sound. Your critique on LLM is also correct. We do not answer questions by remembering all the corpus in the world). However, most people deduce that San Diego is also located further west to Reno, which is unfortunately incorrect. This is an error, but, cognitive psychologists who use them as a window to explore how mind works, precisely, what are the basic representation of mental spatial model, what is the building block, what is the primitive relation, how is mental spatial model used for abstract thinking, and how language structures the mind (what you ask), and how space structures language....
> > > > > >
> > > > > > Hope my writing can be a small amount of money that you get without actual labour.

---

### Decision · Program_Chairs · 2023-09-21

**Decision:**

Reject

**Comment:**

This paper criticizes the limitations of current ML / LM approaches and proposes a new framework. The author argues that no experimental validation is needed, since the proposed framework is 'sound'. The reviewers disagree and don't find this paper fit for NeurIPS. The reviewers engaged with the author and provided some pointers to work that could help the author better situate their approach. In its current form this paper does not seem acceptable to NeurIPS.